# The Role of T Cell Immunotherapy in Acute Myeloid Leukemia

**DOI:** 10.3390/cells10123376

**Published:** 2021-12-01

**Authors:** Fang Hao, Christine Sholy, Chen Wang, Min Cao, Xunlei Kang

**Affiliations:** Center for Precision Medicine, Department of Medicine, University of Missouri, 1 Hospital Drive, Columbia, MI 65212, USA; fhao2@wisc.edu (F.H.); cs6r8@health.missouri.edu (C.S.); wangc1@health.missouri.edu (C.W.); caomi@health.missouri.edu (M.C.)

**Keywords:** acute myeloid leukemia, T cell immunotherapy, T cell alteration, tissue-infiltrated lymphocytes, bispecific antibodies, CAR-T, TCR-T

## Abstract

Acute myeloid leukemia (AML) is a heterogeneous disease associated with various alterations in T cell phenotype and function leading to an abnormal cell population, ultimately leading to immune exhaustion. However, restoration of T cell function allows for the execution of cytotoxic mechanisms against leukemic cells in AML patients. Therefore, long-term disease control, which requires multiple therapeutic approaches, includes those aimed at the re-establishment of cytotoxic T cell activity. AML treatments that harness the power of T lymphocytes against tumor cells have rapidly evolved over the last 3 to 5 years through various stages of preclinical and clinical development. These include tissue-infiltrated lymphocytes (TILs), bispecific antibodies, immune checkpoint inhibitors (ICIs), chimeric antigen receptor T (CAR-T) cell therapy, and tumor-specific T cell receptor gene-transduced T (TCR-T) cells. In this review, these T cell-based immunotherapies and the potential of TILs as a novel antileukemic therapy will be discussed.

## 1. Introduction

Acute myeloid leukemia (AML) is a heterogeneous disease characterized by infiltration of immature myelogenous cells in the blood, bone marrow (BM), and other tissues, which ultimately results in ineffective hematopoiesis and various symptoms of leukemic disease. Because the disease is composed of various cytogenic and molecular abnormalities, individual treatment strategies are complex and require multiple approaches [1,2].

Despite numerous studies aimed at harnessing the power of the natural immunologic antileukemic response, there has been no significant advancement made in the standard treatment regimen in over 50 years [3]. Current treatment consists of a high-intensity induction phase wherein cytotoxic chemotherapy, commonly the “7+3 regimen” of cytarabine plus anthracycline, or other target-specific agents are administered based on disease profile as well as patient risk and comorbidities [3,4,5]. Complete remission (CR) may be attained in approximately 70–80% of patients < 60 years of age or in 50% of patients 60 or older [6]. However, due to frequent residual disease and potential for sequential relapse, a consolidation phase often follows that may include additional chemotherapy, specific antigen-targeting therapies, myeloablative conditioning, or either autologous or allogeneic hematopoietic stem cell transplantation (HSCT) depending on the presence of certain molecular and cytogenetic abnormalities [3,4,6,7].

For those with a poor disease prognosis, allogeneic HSCT is the first-line treatment as it can offer disease cure. It is preferred to autologous HSCT, which demonstrates higher rates of relapse due to various mechanisms of immune escape [8,9,10]. The success of allogeneic HSCT is recognized as a graft-versus-leukemia (GvL) effect, which is mainly attributed to the ability of donor T lymphocytes to target recipient leukemia cells [11]. Donor lymphocyte infusions emphasize the role of T cells in GvL as demonstrated by their effectiveness in relapsed chronic myeloid leukemia (CML) and potential to lower relapse incidence in high-risk AML patients [12,13]. However, despite these advantages, significant side effects of allogeneic HSCT exist, namely, graft-versus-host disease (GvHD), which can be fatal or result in chronic health consequences. Thus, various priming strategies are being investigated to preserve the GvL effect of T cells while avoiding complications encompassing GvHD [10,14,15].

The promising role of T lymphocytes in eradicating leukemic disease has led to the recent development of therapeutics that utilize this cytotoxic potential while minimizing risk for treatment-related morbidity and mortality as well as disease relapse. This review will include a discussion of the various cytotoxic mechanisms and pitfalls demonstrated by T lymphocytes and their implications in the eradication of leukemic cells as well as emerging therapeutics employing these advantageous effects. These will include current treatments aimed at restoring or altering autologous T cell function and cytotoxicity, namely, bispecific antibodies and immune checkpoint inhibitors. Genetically modified T cell immunotherapies, specifically chimeric antigen receptor T (CAR-T) cells and tumor-specific T cell receptor gene-transduced T (TCR-T) cells will be discussed as well. Moreover, tissue-infiltrated lymphocytes (TILs), a treatment strategy demonstrating effective tumor growth inhibition in clinical trials involving various solid malignancies, will be evaluated for their potential role in AML therapy.

## 2. T Cell Alteration in AML

In the presence of AML, multiple clinical studies have demonstrated various disruptions in T cell immunity, including augmented T regulatory cells and reduced T helper cells, T cell exhaustion, and dysregulated activity of transcription factors. To begin with, T cell numbers and functions are altered to allow the progression of myeloid disease. In vitro studies show that coculture with AML cells and incubations with AML plasma inhibit the proliferation of T cells [16,17]. One proven mechanism of T cell growth inhibition is the expression of AML surface receptors that interact with T cells and alter their function. This includes leukocyte immunoglobulin-like receptor B4 (LILRB4), an inhibitory immune checkpoint receptor restrictively expressed on monocytic leukemic cells (M4 and M5 subtypes) (Figure 1). LILRB4 mediates the release of arginase-1, thus directly suppressing T cell proliferation and cytotoxicity in vitro and in vivo [18]. Another immune-suppressive molecule, CD200, is upregulated on the surface of AML cells. CD200 is a type I membrane glycoprotein belonging to the immunoglobulin superfamily whose interaction with the T cell CD200 receptor (CD200R) is associated with both increased regulatory T cell (Treg) populations and reduced memory T cell function [19] (Figure 1). This is one of several mechanisms by which the AML tumor microenvironment may alter subsets of T cells to suppress the immune response. A higher frequency of Tregs impairs the cell-mediated antileukemic immune response [20]. This is supported by an AML mouse model demonstrating an increased frequency of Tregs in vivo in addition to in vitro suppression of effector T cell function. However, with Treg depletion, in vitro effector T cell function was restored, and in vivo treatment outcomes were improved. These results suggest that the expansion of Tregs may be a mechanism of AML persistence [21]. Moreover, increased numbers of Tregs have been identified in AML patients when compared with healthy controls [20]. Shenghui et al. compared the peripheral blood (PB) and BM of 182 patients with newly diagnosed AML with those of 20 healthy age-matched controls [22]. The frequency of Treg cells in PB from patients with AML was significantly higher than that found in the disease-free counterparts (9.2% versus 5.44%, *p* < 0.001). Furthermore, another T cell subset, helper T cells, has demonstrated reduced numbers in AML. T helper 1 (Th1) cells predominantly secrete interferon γ (IFN-γ) and thus play a role in host defense against intracellular pathogens by promoting macrophage activation at sites of inflammation. Th1 cells exhibit a reduced frequency as well as a decreased expression of IFN-γ in AML patients compared with healthy controls (21.03% versus 11.27%, *p* < 0.001) [23]. Patients who obtained CR following standard induction chemotherapy exhibited a significant increase in Th1 cells as opposed to newly diagnosed patients. Additionally, increased frequencies of Th17 cells were identified in the PB and BM of AML patients when compared with healthy donors. Th17 cells secrete interleukin-17 (IL-17), which promotes the proliferation of AML cells and inhibits the differentiation of Th1 cells in vitro. Patients with higher frequencies of Th17 cells had poorer survival than those with lower numbers as well [24,25].

Next, T cell exhaustion is more prominent in AML patients and is demonstrated by an increased number of dysfunctional T cells as well as the expanded expression of inhibitory receptors. Knaus et al. assessed T cell differentiation and function from 72 patients with AML. They identified a significantly higher frequency of terminally differentiated effector cells (CCR7−CD45RA+ and CD27−CD45RA+) with a lower frequency of naïve (CCR7+CD45RA+) and naïve-like cells (CD27+CD45RA+) in patients diagnosed with AML as compared with healthy controls [24]. Moreover, at the time of chemotherapy recovery, the percentages of terminally differentiated cells were significantly higher in PB and BM of nonresponders versus CR patients. Furthermore, other studies have identified the upregulation of inhibitory immune receptors in AML. To illustrate this concept, increased CD8+ T cell populations expressing programmed death 1 (PD-1) and T cell immunoreceptor with Ig and ITIM domains (TIGIT) were identified in AML patients when compared with healthy controls [25,26] (Figure 1). T cells expressing PD-1 and TIGIT were reported to have a significantly reduced ability to produce IFN-γ, tumor necrosis factor alpha (TNF-α), and interleukin 2 (IL-2) in vitro [27]. Additionally, an increased percentage of CD8+ T cell immunoglobulin and mucin-domain containing-3 (TIM-3) expression was reported in the PB of 36 AML patients compared with healthy donors (5.90 ± 4.91 versus 0.96 ± 0.54%) [28]. TIM-3 and PD-1 double-positive T cells isolated from the blood of AML patients and AML murine models were also found to have reduced production of IFN-γ, TNF-α, and IL-2 [29] (Figure 1).

Finally, dysregulation in the T cell expression of certain transcription factors has been established in the pathogenesis of AML. Transcriptomic analysis of CD8+ T cells in AML patient BM revealed the downregulation of *NF-kB*, *Wnt*, and *FoxO*, which are regulative genes required for T cell activation, differentiation, and function. This ultimately leads to disruptions in cytokine and chemokine signaling [30] (Figure 1).

## 3. T Cell Antileukemia Effect

Although T cell immunity is compromised in AML, cytotoxicity against AML cells is still exhibited. An in vitro coculture assay of T cells from AML patients and autologous leukemic cells demonstrated that pooled CD4+ and CD8+ cells were cytotoxic against blasts (32%, 30:1 E/T ratio) [31]. Moreover, Lamble et al. showed an association between T cell numbers and clinical outcomes based on a study involving 66 AML patients. Patients with higher BM T cell percentages (>78.5% of total lymphocytes) were reported to have increased overall survival [32]. Large T cell populations are also associated with leukemia-free survival and the time CR and relapse. The antitumor effect of T cell immunity is mainly credited to CD8+ T cells that recognize mutated peptides expressed and displayed by AML cells and induce cytotoxic effects on these cells. Two studies investigated the role of CD8+ T cells on ridding AML caused by either mutation of nucleophosmin 1 (*NPM1*) or FMS-like tyrosine kinase receptor 3 internal tandem duplication (FLT3-ITD) [33,34]. These represent the most commonly mutated genes found in AML; the presence of NPM1 and FLT3-ITD mutation is reported in 60% and 25% of AML cases, respectively. CD8+ T cells were shown to recognize leukemic blasts expressing such mutations and induce specific lysis of these cells through the secretion of IFN-γ, granzyme B, and perforin. Aside from CD8+ T cells, other studies have reported the cytotoxicity of CD4+ T cells in AML [35,36]. For example, Zhong et al. demonstrated that CD4+ T cell lines exerted proapoptotic effects on AML cells, and the blast apoptosis is correlated with INF-γ secretion [35].

Another T cell subtype, γδ (Vγ9Vδ2) T cells, is becoming an attractive candidate for antileukemic activity [37]. Vγ9Vδ2 T cells, distinct from CD4+ and CD8+ cells, represent a small subset of T lymphocytes found in PB that exhibit potent major histocompatibility complex (MHC)-unrestricted cytotoxicity, high potential for cytokine release, and a broad-spectrum recognition of cancer cells. Gertner-Dardenne et al. confirmed the antileukemic property of Vγ9Vδ2 T cells and demonstrated that Vγ9Vδ2 T cells from AML patients could be isolated and expanded and could recognize and kill AML blasts in vitro and in vivo. They also identified that Vγ9Vδ2 T cells recognized AML blasts primarily via the T cell receptor (TCR) and DNAM-1 receptor and that blasts were killed through a perforin/granzyme-dependent lysis [38].

## 4. T Cell Immunotherapy

Given the effective antileukemic properties of T cells, current immunotherapies aim to restore weakened T cell activity in AML patients through multiple strategies (Figure 2). In this section, we will discuss such therapeutics under clinical investigation for potential AML therapy. These include those that utilize autologous T cells and spare the use of cell modification, including tissue-infiltrating lymphocytes (TILs), bispecific antibodies, and immune checkpoint inhibitors (ICIs). We will also briefly cover the updated information of artificially engineered CAR-T cell therapy and TCR-T cell treatment. Clinical trials of these immunotherapies are summarized in Table 1.

### 4.1. Tissue-Infiltrated Lymphocyte

The use of TIL as a T cell immunotherapy has demonstrated a substantial regression effect in multiple solid cancers, such as melanoma, colorectal, breast, and cervical cancer [39,40,41]. TIL therapy is currently being evaluated in clinical trials in the United States, whereas its approval has been granted in other countries [42]. Although TIL therapy is feasible and efficacious in solid tumors, its potential in treating hematopoietic malignancies, including AML, remains unclear. The existence of TIL in leukemia patients as well as the methodology of TIL isolation from patients requires additional study to clarify the effectiveness of implementing this therapeutic. If applicable to AML treatment, the use of TIL may provide some advantages over other therapeutic options. First, TILs could be isolated from the PB of AML patients without section and lysis of tumor tissue, the latter of which is performed in TIL isolation from solid tumors. Second, unlike CAR-T cell therapy (discussed below), TILs contain innate surface receptors that recognize AML-specific antigens and would not require artificial modification other than amplification, which may save time and spare treatment cost. Following sequence analysis, these TIL surface receptors could aid in the identification of additional AML-specific markers as potential targets for personalized therapy as well as the development of novel treatment strategies. Lastly, TIL involves the use of autologous T cells; thus, cytotoxicity against cancer cells may be maintained while avoiding off-target host tissue destruction.

The lack of TIL research in AML treatment may be attributed to the disruptions in T cell function that result from leukemic disease [43]. As discussed previously, numerous AML patient studies have identified an augmented population of Tregs, diminished helper T cell function, reduced IFN-γ secretion, and a significantly increased frequency of terminally differentiated or exhausted CD8+ T cells compared with healthy controls [23,24,44]. Although these observations indicate poor T cell populations and dysfunctional effector action, the phenotype and cytotoxic functions of TILs in AML require further elucidation.

Two studies recently demonstrated reasonable solutions to isolate TILs in AML [45,46]. Wei et al. isolated TILs from the BM of 20 patients via CD3+ staining and immunomagnetic selection and found that TILs were detected in 50% of the patient samples [45]. They also expanded TILs in cocultures with irradiated feeder cells supplemented with IL-7 and IL-15. Expanded TILs were noted to secrete IFN-γ (mean of 2228.5 pg/mL) and display a significantly higher antitumor response against the K562 cell line compared with that of autologous PB lymphocytes. Moreover, García-Guerrero et al. successfully isolated antigen-specific T cells (doublet-forming T cells) using FACS-based cell sorting and first identified the doublet formation of TILs plus target cells under the microscope. They incubated peripheral blood mononuclear cells (PBMCs) from patients who achieved CR with autologous AML cells and selected doublet-forming T cells that bonded with AML cells; these cells expressed the CD3 T cell receptor as well as an AML cell marker simultaneously [46]. They found that doublet-forming T cells display a higher percentage of CD8+ T cells, increased IFN-γ secretion, and upregulated expression of the T cell activation marker CD69 and the cytotoxic marker CD107a. Doublet-forming T cells also exhibited marked cytotoxic activity against autologous AML blasts in vitro. Hence, García-Guerrero et al. provide a feasible approach of isolating tumor-specific and cytotoxic TILs from AML patients without the heterogeneity of a T cell population in conventional isolation methods in terms of IFN-γ production and MHC affinity. Nevertheless, the studies discussed have limitations. First, the doublet-forming T cells selected in this study, which exhibit both CD3 T cell and AML cell markers as well as a relatively large cell size, might be individual T cells and AML cells in a close position in place of actual TILs. Second, this study did not test the antileukemic effect of doublet-forming T cells in vivo, so the clinical implication and value of doublet-forming T cells are deficient. Third, the presence of T cells adhesive to AML cells still needs to be rectified by additional research in newly diagnosed patients since the doublet-forming T cells in this study were isolated from patients completing chemotherapy or who had achieved CR.

Regarding the maintenance of T cell populations, another study conducted by Beltra et al. identified subsets of CD8+ T cells undergoing exhaustion and transcription factors involved in the progression [47]. Through the use of a mouse model under chronic viral infection, they defined four subsets of exhausted CD8+ T cells (Tex) according to differential expression levels of CD69 and Ly108 (*Slamf6*). These were progenitor 1 (Ly108+ CD69+), progenitor 2 (Ly108+ CD69-), intermediate (Ly108− CD69−), and terminal (Ly108− CD69+). The study found that compared with progenitor stages, T cells in intermediate and terminal stages lose ex vivo proliferative potential and exhibit lower levels of IFN-γ in addition to higher levels of caspase-3. Through analysis of protein-level dynamics among these four stages of T cells, they also showed that the expression of two transcription factors, TCF-1 and T-bet, decreases gradually from the progenitor stage to the terminal stage. Furthermore, the expression of Tox, a high-mobility group (HMG) protein, was associated with the terminal stage. CD8+ T cells with Tox knockout displayed higher levels of T-bet and increased proliferation. Therefore, the study concluded that the expression of TCF-1 and T-bet favors the accumulation of Tex in progenitor or intermediate stages and reduces the number of terminally exhausted cells, the latter of which is antagonized by Tox expression. In short, Beltra et al. established methods of monitoring the exhaustion status of T cells and reversing T cell exhaustion, thus providing helpful clues for the maintenance of TILs isolated from AML patients.

We conclude that there are several directions for future TIL research in AML. It is critical to investigate and confirm the suppressive effects of doublet-forming T cells against AML based on in vivo models and patient samples. Additionally, the exhaustion subsets of the CD8+ TILs in AML, based on CD69 and Ly108 levels, need further clarification. Importantly, future researchers should identify whether TCF-1 and T-bet overexpression or Tox inhibition can rescue T cell exhaustion in AML patients and promote cell-mediated cytotoxicity against AML cells.

### 4.2. T Cell-Recruiting Bispecific Antibody

The clinical role of antibody-based therapy currently approved for AML treatment is limited to antibody-drug conjugates (ADCs), wherein an innate immune response is elicited towards a myeloid-specific antigen, such as CD33-binding gemtuzumab ozogamicin [5,48,49]. However, several antibody-based therapies aimed at inducing a cell-mediated cytotoxic response against leukemic cells are being investigated for AML therapy. Collectively referred to as bispecific antibodies, these are constructs capable of synchronously recognizing and binding two distinct antigens, including one tumor antigen plus another of an effector immune cell, most commonly the T cell receptor CD3. The yield is the recruitment of effector T cells towards targeting blast cells independent of natural TCR–MHC activation mechanisms, which provides the highly advantageous avoidance of immune escape via tumor cell MHC downregulation [50,51,52]. Some bispecific antibodies, such as bispecific T cell engagers (BiTEs), may be implicated as a strategy to reactivate exhausted T cells as well [53]. Similar constructs exist, such as tandem diabodies and dual-affinity retargeting antibodies (DARTs). Today, blinatumomab, a BiTE that binds to CD3 on T cells and CD19 on B cells, is one of the few FDA-approved bispecific antibodies and is indicated for the treatment of relapsed or refractory B cell progenitor acute lymphoblastic leukemia (BCP-ALL). The success of blinatumomab has led to the booming interest in and development of various bispecific antibody constructs linking T cells to myeloid antigens and thus providing novel AML therapeutics [52,53,54].

CD33 (Siglec-3) is a 67 kDa transmembrane glycoprotein that is expressed on multipotent myeloid precursors as well as maturing granulocytes and monocytes, although a low level has been detected on subsets of B cells, activated T cells, and NK cells. It is widely expressed in AML on leukemic stem and progenitor cells; however, CD33 is either absent or expressed at low levels in normal hematopoietic stem cells [52,55,56]. In fact, CD33 expression is in approximately 85%–90% of AML patients [57]. As mentioned previously, gemtuzumab ozogamicin, an anti-CD33 ADC delivering cytotoxic therapy to leukemic cells, has proven to be effective in cases of AML in conjunction with various treatment regimens [58]. Thus, immunotherapeutic strategies targeting CD33 are of high interest and have been implemented in AML clinical trials.

AMG 330 (NCT02520427) is a short-acting human CD3/CD33 BiTE that has demonstrated increased T cell recruitment, expansion, and antibody-mediated cytotoxicity, resulting in the depletion of autologous blasts in ex vivo studies. These effects were achieved in low E-to-T ratios, mimicking the phenotype of patients with active disease and thus providing further support for the use of AMG 330 in vivo. Among T cell activation markers and cytotoxic cytokine secretion increases observed were LAG3, TIM3, and PD-1 and INF-γ, TNF, IL-2, and IL17A, respectively [59]. AMG 330 is currently in phase I clinical trial for adult R/R AML and is administered as continuous IV doses varying and up to 720 µg/day. At this time, 8 out of 42 evaluated patients demonstrated responses, of which 3 were CRs, 4 CR with incomplete hematologic recovery (CRi), and 1 morphologic leukemia-free state. The most common adverse event observed was cytokine release syndrome (CRS), which was noted to be reversible and occurred in a dose-dependent manner [60]. In addition, ex vivo studies demonstrated increased expression of PD-L1 on AML cells with AMG 330 treatment. With the addition of a PD-1/PD-L1-blocking antibody, this study demonstrated AMG 330-mediated increased T cell proliferation, INF-γ secretion, and AML cell lysis. The use of immune checkpoint inhibitors with AMG 330 has not been indicated in clinical trials; however, these findings support the potential benefit of combination therapy [61].

In contrast with AMG 330, AMG 673 (NCT03224819) is a half-life extended BiTE consisting of the N-terminus of a single-chain IgG Fc region plus CD3/CD33 binding sites currently in phase I clinical trial for adult R/R AML. AMG 673 appears advantageous over AMG 330 as it does not require continuous infusion due to its extended half-life. Instead, the BiTE is administered as two IV infusions per 14-day cycle, the median cycle repetition being 1.5. The last update of AMG 673 provided notes 30 treated patients with 67% having 4 or more previous leukemic treatments and 23% having received HSCT prior to enrollment. Of the patients analyzed, 44% demonstrated blast reduction. One patient achieved CRi with an 85% reduction of BM blasts, while 22% of the patients achieved a reduction of at least 50%. Similar to AMG 330, the most common adverse effect observed was CRS. It has been reported that doses up to 72 µg may induce blast reduction while limiting adverse effects [62]. The trial is estimated to complete in December 2021.

Another phase I clinical trial for adult R/R AML is AMV564 (NCT03144245), a bivalent, bispecific CD3/CD33 T cell engager whose construct allows an extended half-life. The trial design includes 14 consecutive days of continuous AMV564 intravenous infusion occurring in 28-day cycles. Out of 35 evaluable patients participating in the trial, 49% demonstrated reductions in BM blasts, including 1 CR, 1 CRi, and 1 partial response (PR). Notably, BM T cells increased with sequential treatment cycles. Furthermore, 3 patients demonstrated increased serum neutrophil counts [63]. Ex vivo studies confirmed little or absent cytotoxicity against differentiated myeloid cells, including neutrophils and monocytes [64]. Lastly, another experimental arm of AMV564 is in combination with pembrolizumab, a PD-1 inhibitor, and the results have yet to be announced.

Other bispecific CD3/CD33-targeting antibodies in phase I clinical trials for adult R/R AML include JNJ-67571244 (NCT03915379) and GEM333 (NCT03516760). Clinical testing of these began more recently, and significant results are to be revealed.

Additional myeloid targets are currently under clinical trial investigation for AML bispecific antibody therapy as well. These include CD123 and CD135, and the final results are to be anticipated [60,62].

### 4.3. Immune Checkpoint Inhibitor

Another strategy to restore compromised T cell immunity is the ICI. As discussed previously, T cell expression of immune checkpoint receptors, such as PD-1 and TIGIT, is upregulated in AML. Current ICIs for AML treatment are predominantly anti-PD-1 antibodies, such as nivolumab, ipilimumab, and pembrolizumab. Combination treatments between ICIs and chemotherapy drugs have undergone clinical trials and shown to suppress AML progression. In one phase II study (NCT02397720), nivolumab, combined with azacitidine, was administered to 70 AML patients, yielding a response rate of 33%, including 22% with CR [65]. Furthermore, 14% of the patients achieved CR and had longer overall survival (OS) after receiving pembrolizumab with azacitidine. The combination of nivolumab and pembrolizumab with cytarabine resulted in more favorable clinical outcomes [66] (NCT02845297). According to other phase II studies (NCT04214249, NCT02464657), the CRs of patients who received the combination therapy were 64% and 38%, respectively [67,68]. Despite the benefits in clinical outcomes demonstrated by the use of ICIs, treatment-related toxicities resulting from dysregulation in the immune system balance hampered the approval of these therapeutics [69]. Patients receiving ICIs were at a higher risk of infections. Grade 3/4 immune-mediated toxicities, such as pneumonitis, rash, and elevated serum bilirubin, were seen in several patients receiving nivolumab or pembrolizumab [70]. In addition, long-term implications of ICI use include resistance [71]. Adaptive resistance to ICIs was observed across multiple studies through compensatory upregulation of alternative immune checkpoint receptors, including TIGIT, TIM-3, and LAG-3. Furthermore, an increased presence of myeloid-derived suppressor cells (MDSCs) and tumor-associated macrophages as well as other immunosuppressive factors in the tumor microenvironment, such as TGF-β and VEGF, may serve to circumvent PD-1 blockade and promote immune evasion and tumor growth.

### 4.4. Chimeric Antigen Receptor T (CAR-T) Cell Therapy

Unlike bispecific antibodies and ICIs that transiently direct and promote patient T cells against AML cells expressing specific target ligands, CAR-T cells are genetically engineered autologous T cells equipped with a synthetic target antigen receptor (CAR) and expand after transfusion in a target antigen-dependent matter [54]. The binding between a CAR and its leukemic cell ligand triggers a signal transduction cascade that ultimately activates target killing of T cells.

Despite the advantageous antileukemic effects of CAR-T cell therapy, AML antigens targeted by CAR-T cells are frequently expressed in normal hematopoietic cells or healthy organ tissues, and the use of these therapeutics can result in on-target, off-tumor toxicities, thus leading to the destruction of various host tissues. Accordingly, most clinical trials are currently applying CAR-T cell therapy as a “bridge to transplant” strategy, with the aim of eradication of residual AML cells to reduce relapse rates following allogeneic HSCT. In accordance with bispecific antibodies, current CAR-T cells in clinical trials predominantly target CD33, CD123, and CLL-1. Most CAR-T cell therapies are in preclinical development or just beginning to enter clinical application in phase I trials. One clinical trial involving six patients showed that all participants receiving CD123 CAR-T cells had refractory AML following allogeneic HSCT [72] (NCT02159495). Two patients had blast reduction, and one achieved a morphologic leukemia-free state after the second allogeneic HSCT. There are also two clinical studies testing anti-CD33 and anti-Lewis Y CAR-T cells, but all patients relapsed after 1–23 months [73,74] (NCT01864902). Although results from most studies have not yet been published, a review of the results from conference proceedings suggests that the minority of patients achieved lasting CR. Frequent relapse occurrence in these clinical trials suggests several limitations of CAR-T cell therapy in AML [75]. First, CAR-T cell therapy requires highly advanced technology and vast knowledge regarding the therapeutic technique; this can only be offered in specialized centers where AML is also a research focus. Second, according to clinical trials, there are several adverse effects associated with CAR-T cell treatment. These include but are not limited to tumor lysis syndrome, CRS, and prolonged cytopenia. Finally, current CAR-T cell strategies do not address the complexity of AML and AML leukemic stem cells explicitly, but rather focus on only one or a few target structures.

### 4.5. Tumor-Specific T Cell Receptor Gene-Transduced T Cells

As discussed earlier, one benefit of TIL investigation is providing the framework for TCR-T development. Unlike CAR-T cells that have receptors targeting naturally occurring antigens, TCR-T cells are engineered to express a transgenic T cell receptor specific for a tumor-associated antigen formed only from the complex of a tumor peptide and an MHC molecule [76]. To determine reactivity to a specific target, binding and functionality are assessed by measuring MHC multimer binding and immune reaction via ELISA or flow cytometry. Once responding T cells are detected, the TCR sequence is determined and transferred to a new T cell. Current TCR-T cells in clinical trials are established based on overexpressed antigens derived from wild-type proteins with relatively high expression in malignant cells. The Chapuis and Greenberg group engineered and expanded donor-derived CD8+ T cells specific for Wilms’ tumor 1 (WT1) epitope ex vivo [77,78] (NCT01640301, NCT00052520). WT1 is overexpressed in acute leukemias and myelodysplastic syndromes (MDS) and has limited expression on normal CD34+ hematopoietic stem cells. They found that WT1-specific T cells supplemented with IL-21 delayed AML relapse for up to 21 months in 11 patients after HSCT, compared with relapse at 16 months among matched controls. However, no survival advantage over standard care was proven in patients treated with WT1 TCR-T cells in preliminary findings. The Tawara group also developed a TCR-T with a retroviral construct encoding WT1 [79]. Eight patients with chemotherapy-refractory AML or high-risk MDS were treated with WT1-specific T cells. Although patients with TCR-T treatment showed no survival advantage, they found that WT1 TCR-T cells persisted for 3 months and displayed a transient decrease in BM leukemic blasts in five patients. Moreover, these studies demonstrated that the transferred T cells were well tolerated with minimal on-target, off-tumor toxicity. The data suggest that transferred WT1 TCR-T cells are well tolerated and may have antileukemic activity in vivo.

Furthermore, several ongoing clinical trials are investigating TCR-T specific to WT1 (NCT00620633, NCT01758328) as well as preferentially expressed antigen in melanoma (PRAME) (NCT02743611, NCT03503968), both of which have exhibited overexpression patterns in solid tumors, ALL, AML, and MDS [80,81].

## 5. Conclusions

In conclusion, significant progress has been made regarding understanding the complexity of the immune biology of AML in addition to noteworthy advancements in the technical development of novel T cell-based AML therapeutics. Despite numerous ongoing trials, T cell immunotherapies for myeloid malignancies remain at an elementary stage. We predict that clinical advancements in immunotherapy will be facilitated by an increased focus on the analysis of potential biomarkers as therapeutic targets. This will aid in elucidating the mechanisms of immune resistance, predicting patient response, and allow for improved management of undesired toxicities. Other than biomarker-driven approaches, the application of novel genomic and cellular techniques, such as single-cell genome sequencing, single-cell cytokine analysis, and mass cytometry, using patient samples will help to reveal additional T cell roles in the tumor microenvironment. These methods may also lead to the discovery of immune components other than cell-mediated that contribute to the immune response as well as escape, ultimately guiding the development of innovative immune therapy strategies. To illustrate, single-cell TCR sequencing generates a TCR repertoire encompassing a wide array of T cell populations in terms of antigen specificities. T cells expressing specific receptors that recognize MHC on tumor cells are likely to characterize TILs. These TCR sequences may identify TIL markers to be used for TIL isolation and amplification. The sequences generated can also be used to identify novel target antigens for personalized immunotherapy and vaccines. Thus, in the near future, we expect promising clinical outcomes for T cell immunotherapies in AML.

## Figures and Tables

**Figure 1 cells-10-03376-f001:**
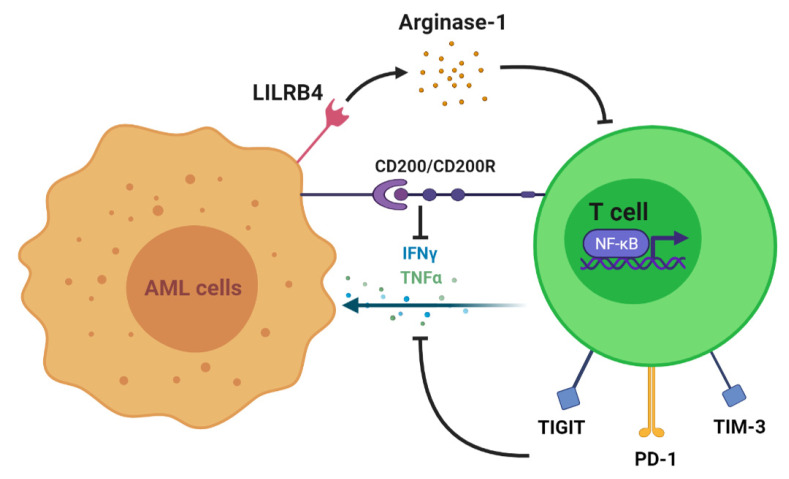
Mechanisms of T cell suppression and dysfunction induced by AML. LILRB4 is an inhibitory receptor present in monocytic AML that directly suppresses T cell proliferation. T cell proliferation and activation are also hindered due to the downregulation of transcription factors, including NFκB, by the tumor microenvironment. Upregulated expressions of checkpoint molecules, such as Tim-3, PD-1, and TIGIT, on T cells and interactions between CD200 and CD200R reduce the secretion of IFN-γ and TNF-α, thus promoting immune evasion.

**Figure 2 cells-10-03376-f002:**
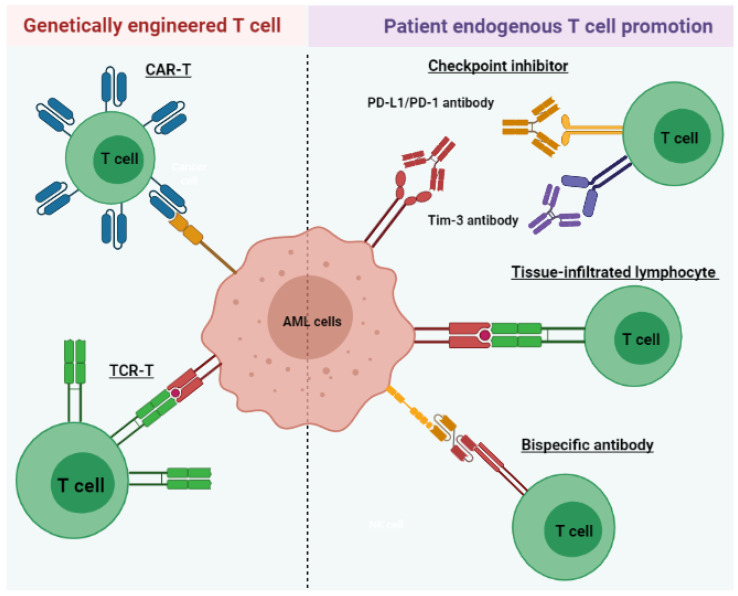
Immunotherapeutic strategies include the promotion of endogenous patient T cells and genetically engineered T cells for AML treatment. Checkpoint inhibitors prevent immune-suppressive signaling through PD-1/PD-L1 and Tim-3. TILs express surface receptors (green) that specifically target AML antigens and exhibit increased cytotoxicity. Bispecific antibodies bind and cross-link leukemic antigens to T cells, facilitating their antileukemic activities. CAR-T or TCR-T provides another strategy to improve leukemia-directed T cell activity via genetic modification of the T cell surface. TCR-T cells express receptors specific for a tumor-associated antigen presented in MHC molecules, similar to TIL. CAR-T cells express receptors (blue) targeting nonspecific, naturally occurring antigens.

**Table 1 cells-10-03376-t001:** Clinical trial information of T cell immunotherapies.

Name	Target CheckPoint/Antigen	Clinical Trial Start Time	Trial Number/Phase	Disease	Status
T cell-recruiting bispecific antibody	CD3 and CD33	2015	NCT02520427/Phase 1	Relapsed/Refractory AML	Recruiting; estimated completion on 28 February 2023
		2017	NCT03224819/Phase 1	AML	Active; estimated completion on 21 March 2022
		2017	NCT03144245/Phase 1	AML	Completed on 5 November 2020; no result posted
		2019	NCT03915379/Phase 1	AML and myelodysplastic syndromes	Recruiting; estimated completion on 26 October 2022
		2018	NCT03516760//Phase 1	Relapsed/Refractory AML	Active; estimated completion on 31 December 2020
Immune checkpoint inhibitor	PD-1	2015	NCT02397720/Phase 2	AML	Recruiting; estimated completion on 30 April 2022
		2016	NCT02845297/Phase 2	AML	Active; estimated completion in October 2021
		2020	NCT04214249/Phase 2	AML	Not yet recruiting; estimated completion on 31 July 2024
Chimeric antigen receptor T cell therapy	CD123	2015	NCT02159495/Phase 1	AML	Recruiting; estimated completion on 15 December 2021
	CD33/Lewis Y	2013	NCT01864902/Phase 1 and 2	Relapsed/refractory AML	Unknown recruiting status; estimated completion in April 2017
Tumor-specific T cell receptor gene-transduced T cells	WT 1	2012	NCT01640301/Phase 1 and 2	Recurrent and secondary AML	Active; estimated completion on 1 September 2029
		2002	NCT00052520/Phase 1 and 2	AML	Completed in June 2013; no result posted

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
