# Peer review of "The Role of T Cell Immunotherapy in Acute Myeloid Leukemia"

_cells, 2021, doi:10.3390/cells10123376_

Round 1

Reviewer 1 Report

The authors provide a well-organized and well-described review of the role of the role of T cell immunotherapy in acute myeloid leukemia. However, authors should explain or justify better what their review contributes to the other reviews already published.

Author Response

Our review, compared to others, includes more information about tissue infiltrating lymphocyte (TIL) as a new potential immunotherapy for AML that has not been thoroughly investigated yet. We demonstrate advantages of TIL therapy and discuss previous studies regarding TIL amplification and maintenance. Given that other reviews provide more emphasis on immune checkpoint and CAR-T cell therapies, our review complements understanding and contributes to the broad picture of AML immunotherapy from another aspect.

Reviewer 2 Report

This is an intriguing paper regarding on-going efforts to employ T cells to improve immunotherapy outcome in AML. 

  1. Line 41: The role of autologous stem cell transplant is very limited in AML in these days and the decision to perform allogenic stem cell transplantation is based on molecular and cytogenetic abnormalities, rather than the amount of residual disease. The residual disease in this paragraph can be misleading so please rephrase these two points.
  2. Line 73: LILRB4 is distinctly expressed on monocytic leukemia (M4 and M5), which should be clearly described in the text. 
  3.  Line 126: mucin-domaining containing-3 --> T cell immunoglobulin and mucin-domain containing-3 (TIM3)
  4. Line 134: NF-kB, Wnt, and FoxO regulated genes are downregulated, not the FoxO gene itself, so please clarify.
  5. Line 187: given the that --> given that
  6. Line 188: myeloid blasts are primarily located in PB --> BM
  7. Line 193: as well as further the development --> as well as the development
  8. Line 202: please check the space bar
  9. Line 226: please discuss that CD3 is also expressed on the subset of AML cells
  10. Line 229: controlling exhaustion development --> controlling exhaustion
  11. Line 296: PDL-1 --> PD-L1
  12. Line 383: Tumor lysis syndrome and CRS are not on-target/off-tumor effects. Please rectify.
  13. Line 437: Please also discuss the efforts regarding single-cell TCR sequencing, in addition to the single-cell RNA sequencing

  1.  

Author Response

We revised each problem accordingly. Each revised part is marked in the updated draft.

  • Line 41: The role of autologous stem cell transplant is very limited in AML in these days and the decision to perform allogenic stem cell transplantation is based on molecular and cytogenetic abnormalities, rather than the amount of residual disease. The residual disease in this paragraph can be misleading so please rephrase these two points.

We remove the amount of residual disease as the dependent factor of allo-HSCT.

  • Line 73: LILRB4 is distinctly expressed on monocytic leukemia (M4 and M5), which should be clearly described in the text. 

We added specific FAB types of AML with LILRB4 expression.

  • Line 126: mucin-domaining containing-3 --> T cell immunoglobulin and mucin-domain containing-3 (TIM3)
  • Line 134: NF-kB, Wnt, and FoxO regulated genes are downregulated, not the FoxO gene itself, so please clarify.

We clarified that NF-KB, Wnt, and FoxO were all regulated genes are downregulated.

  • Line 187: given the that --> given that
  • Line 188: myeloid blasts are primarily located in PB --> BM

We rephrase our idea about TIL localization. We would like to emphasize that TILs, in the case of AML, can be isolated from PB samples of patients.

  • Line 193: as well as further the development --> as well as the development
  • Line 202: please check the space bar
  • Line 226: please discuss that CD3 is also expressed on the subset of AML cells

We discussed a bit more about the property of doublet-forming cells expressing both CD3 and AML cell marker.

  • Line 229: controlling exhaustion development --> controlling exhaustion
  • Line 296: PDL-1 --> PD-L1
  • Line 383: Tumor lysis syndrome and CRS are not on-target/off-tumor effects. Please rectify.

We clarified tumor lysis syndrome and CRS as adverse effects of CAR-T treatment.

  • Line 437: Please also discuss the efforts regarding single-cell TCR sequencing, in addition to the single-cell RNA sequencing

We discussed more about how single-cell TCR-T sequencing facilitates the identification of TIL population and recognition of AML-specific antigens for future development of targeting therapies.

Reviewer 3 Report

The authors report the role of T cell immunotherapy in acute myeloid leukemia.

  1. The authors should summarize the clinical trial list in a Table.

Author Response

We summarize information of clinical trials in Table 1 at the end of the article.